# Soluble Guanylyl Cyclase Activators—Promising Therapeutic Option in the Pharmacotherapy of Heart Failure and Pulmonary Hypertension

**DOI:** 10.3390/molecules28020861

**Published:** 2023-01-15

**Authors:** Grzegorz Grześk, Adrianna Witczyńska, Magdalena Węglarz, Łukasz Wołowiec, Jacek Nowaczyk, Elżbieta Grześk, Alicja Nowaczyk

**Affiliations:** 1Department of Cardiology and Clinical Pharmacology, Faculty of Health Sciences, Ludwik Rydygier Colle-gium Medicum in Bydgoszcz, Nicolaus Copernicus University, 75 Ujejskiego St., 85-168 Bydgoszcz, Poland; 2Department of Organic Chemistry, Faculty of Pharmacy, Ludwik Rydygier Collegium Medicum in Byd-goszcz, Nicolaus Copernicus University in Toruń, 2 dr. A. Jurasza St., 85-094 Bydgoszcz, Poland; 3Physical Chemistry and Chemistry of Polymers, Faculty of Chemistry, Nicolaus Copernicus University, 7 Gagarina St., 87-100 Toruń, Poland; 4Department of Pediatrics, Hematology and Oncology, Faculty of Medicine, Ludwik Rydygier Collegium Medicum in Bydgoszcz, Nicolaus Copernicus University, 9 Skłodowskiej-Curie St., 85-094 Bydgoszcz, Poland

**Keywords:** guanylate cyclase (GC), chronic heart failure (CHF), pulmonary arterial hypertension (PAH), vericiguat, riociguat

## Abstract

Endogenous nitric oxide (NO)-dependent vascular relaxation plays a leading role in the homeostasis of the cardiovascular, pulmonary, and vascular systems and organs, such as the kidneys, brain, and liver. The mechanism of the intracellular action of NO in blood vessels involves the stimulation of the activity of the soluble cytosolic form of guanylyl cyclase (soluble guanylyl cyclase, sGC), increasing the level of cyclic 3′-5′—guanosine monophosphate (cGMP) in smooth muscle and subsequent vasodilation. In recent years, a new group of drugs, soluble guanylyl cyclase stimulators, has found its way into clinical practice. Based on the CHEST-1 and PATENT-1 trials, riociguat was introduced into clinical practice for treating chronic thromboembolic pulmonary hypertension (CTEPH). In January 2021, the FDA approved the use of another drug, vericiguat, for the treatment of heart failure.

## 1. Introduction

In 2012, at the annual meeting of the American College of Chest Physicians (ACCP) in Atlanta, the results of two trials, CHEST-1 and PATENT-1, were presented, confirming the efficacy of riociguat in the treatment of chronic thromboembolic pulmonary hypertension (CTEPH) [1,2]. In addition, in 2021, the FDA approved the use of vericiguat for the treatment of heart failure. Research into the use of other sGC activators—including cinaciguat or pralinciguat—is still ongoing.

Nitric Oxide Pathway as a Target for Pharmacological Intervention in Pulmonary Arterial Hypertension and Heart Failure

There is a common element in the pathogenesis of diseases, such as heart failure and pulmonary arterial hypertension. It is the process of the hypertrophy and remodeling of the vascular wall coexisting with progressive fibrosis. The dysfunction of the vascular endothelium is of great importance in this process. The pharmacotherapy of both diseases is aimed at intervening in the pathways involved in the influence of mediators [3].

The impact of nitric oxide donors is short-lived due to the time-limited possibilities of the enzymatic release of nitric oxide, as well as the toxicity of some drugs. For this reason, research on the next links in the mechanism of nitric oxide or other pathways leading to an increase in the concentration of the key link—cGMP—has become so important [4,5].

The modern pharmacotherapy of heart failure includes converting enzyme inhibitors or neprilysin inhibitors, mineralocorticoid receptor antagonists, beta-adrenoreceptor antagonists, and sodium–glucose transport protein type 2 inhibitors. Among the drugs included in the next steps are also guanylate cyclase activators. The pharmacotherapy of pulmonary arterial hypertension includes prostacyclin analogs, phosphodiesterase-5 inhibitors, endothelin-1 receptor antagonists, and a modulator of guanylate cyclase activity. During the therapeutic effect of the above-mentioned groups of drugs, an increase in cGMP concentration is obtained directly or indirectly [3,4,5]. Angiotensin converting enzyme inhibitors, by inhibiting the degradation of bradykinin, increase its concentration. A similar effect occurs with the use of neprilysin inhibitors, but in addition to inhibiting the degradation of bradykinin, the degradation of natriuretic peptides is also inhibited. Phosphodiesterase-5 inhibitors inhibit cGMP degradation. In the case of beta-adrenoceptor antagonists, mineralocorticoid receptors, or sodium–glucose inhibitors of protein transport type 2, there is also an indirect relationship [5,6,7].

The idea of the direct modulation or stimulation of guanylate cyclase seemed promising from a theoretical point of view; moreover, clinical trials confirmed the efficacy in these clinical settings. The main goal was to increase the activity of the enzyme and thus achieve a synergistic effect between the applied pharmacotherapy and the activation of guanylate cyclase [5,8,9].

## 2. The Action of sGC Activators

Nitric oxide (NO) is an endogenous vasodilator synthesized by nitric oxide synthase (eNOS, NOS III, and NOS3) and released constantly from endothelial cells. Nitric oxide synthase exists in three isoforms, named after the tissue from which the enzyme was isolated initially: neuronal (NOS I, also called nNOS, NOS1), endothelial (eNOS, NOS3, NOS III), and inducible (iNOS, NOS2 NOS II). NOS1 is responsible for the transmission of nerve impulses, while NOS3 plays a role mainly in the vasodilation and regulation of arterial pressure [3].

NO signaling has pleiotropic roles In biology and a crucial function in cardiovascular homeostasis. An increase in NO secretion occurs under the influence of mediators, such as norepinephrine (NA), angiotensin, adenosine triphosphate, or bradykinin. NO synthesis is also stimulated as a result of numerous physical factors., e.g., increasing mechanical pressure and the pressure exerted on the endothelium by the flowing bloodstream or low oxygen partial pressure in the blood vessels [3,4,5]. 

The mechanism of NO’s intracellular action is primarily through the stimulation of the activity of the soluble form of guanylyl cyclase (sGC). The sGC is heme-containing enzyme, which causes an increase in the level of cyclic 3′-5′-guanosine monophosphate (cGMP) in smooth muscle and subsequent vascular relaxation [6]. For years, guanylate cyclase was thought to be a homogeneous, tissue-unspecific enzyme, but studies have shown that there are two main types of guanylate cyclase, which have different localization within the cell. The first is a membrane-bound cyclase (mGC), while the second is an entirely intracellular soluble guanylate cyclase (sGC). For the former, the agonists are peptides (natriuretic peptides A, B, C), and for the latter, they are gaseous mediators (nitric oxide, carbon monoxide).

NO/sGC/cGMP regulation plays a leading role in the homeostasis of the cardiovascular and pulmonary systems and organs, such as the kidney, brain, and liver. In addition to smooth muscle cells, cGMP influences the function of fibroblasts, cardiomyocytes, platelets, neurons, and immune cells, regulating the processes of fibrosis, inflammatory response, and neurotransmission [3,4,5,7] (Figure 1).

The discovery of new classes of compounds, the so called sGC stimulators and sGC activators, or in medical lingo guates, that stimulate cGMP formation provided a tool to study sGC redox regulation and its role in pathogenesis. It also made it possible to target drugs directly at diseased blood vessels, heart muscle, kidneys, and other organs. From a pharmacological point of view, sGC stimulators increase the activity of sGC independently of NO and also act synergistically with endogenous NO. In contrast, sGC activators specifically bind to and activate an oxidized, heme-free form of sGC [8,9] [Table 1].

The sGC stimulators described to date, including lificiguat, riociguat, ataciguat, and cinaciguat, bind directly to sCG. In addition, riociguat was found to sensitize sGC to endogenous nitric oxide by stabilizing the nitric oxide–sGC bond [10,11,12].

## 3. Molecular Structures of the sCG Active Compounds

The molecular formulas of the sCG active compounds are gathered in Figure 2.

The first discovered sGC active compound was lificiguat (YC-1). From a chemical point of view, lificiguat is an indazole derivative whose structure contains furyl and benzyl substituents at positions 1 and 3, respectively. This benzylindazole compound does not exist in the nature and is obtained only by chemical synthesis [13] (Figure 2). The class of substances that was developed based on the modification of the structure of lificiguat originated from nelociguat. This group of substances includes pyridine pyrimidinopyrazole derivatives as the leading structure [13] Nelociguat can be classified as pyrazolopyridine, aminopyrimidine, organofluorine, and carbamate ester. The addition of one methyl group to the carbamate ester moiety results in a riociguat structure. Riociguat is the first compound in the sGC activator class approved as a drug [14] (Figure 2). It is worth noting that vericiguat (sGC activators, Figure 2) is an analog of the structure of nelociguat. The vericiguat structure has one additional fluoro substituent on the 1H-pyrazolo [3,4-b] pyridine moiety. SAR studies revealed that the introduction of an additional fluorine atom in the nelociguat changed the nature of the activity of these molecules from the stimulators to the activator of sGC [15]. 

Ataciguat is the anthranilic acid derivatives member. It stimulated sGC in a concentration-dependent and reversible fashion (EC50 of 0.51 μM and was active in vascular smooth muscle cells exposed to oxidative stress) [16]. Cinaciguat and BAY 60-2770 are examples of compounds that prove that an active sGC compound cannot contain a heterocyclic ring in the structure (Figure 2). Praliciguat, oliciguat, and vericiguat are shown as examples of sGC activators.

Apart from efficacy and toxicity, many drug development failures are imputable to poor pharmacokinetics and bioavailability. Bioavailability can be described by six physicochemical properties indices, such as lipophilicity, size, polarity, solubility, flexibility and saturation. [17]. Figure 3 shows the assessment of the similarity of drugs of the discussed compounds to known drugs in the form of bioavailability radar [18]. The presented comparison shows that the basic properties of the stimulators/activators synthesized so far differ from the known drugs [19]. This indicates some complex bioavailability process and the need for the careful selection of the dose of the preparations used.

Of the various routes of drug administration, oral dosing of the drug is highly preferred. Therefore, the early estimation of oral bioavailability and the fraction of the dose that reaches the blood after oral administration is crucial in the drug discovery process. Nevertheless, bioavailability is highly multifactorial but depends mainly on absorption from the gastrointestinal tract, including metabolism in the gut wall [20]. The routinely applied prediction tool is the Brain Or IntestinaL EstimatED permeation method (BOILED-Egg, also called the Egan egg) proposed by Egan et al. [21] illustrate the gastrointestinal absorption (HIA) and brain penetration (BBB). The pharmacokinetic properties of the discussed compounds using the boiled egg model allows for the intuitive evaluation of passive gastrointestinal HIA and BBB in the function of the position of the molecules in the Log *P* versus TPSA referential (Figure 4). BOILED-EGG is fast and acceptable as an accurate predictive model (with an internal accuracy above 93 %) for pharmacokinetics and bioavailability drug behaviors. The white region is for high probability of passive absorption by the gastrointestinal tract, and the yellow region (yolk) is for high probability of brain penetration. Yolk and white areas are not mutually exclusive [22].

As shown in Figure 4, the values of the obtained physicochemical parameters for lificiguat indicate a high probability of penetration into the brain. In contrast, BAY 60-2770, cinaciguat, riociguat, and praliciguat have a high probability of penetrating the blood from the gastrointestinal tract. Values below the optimal probability of good bioavailability can be attributed to oliciguat, verciguat, nelociguat, and ataciguat. In addition, we can expect that all analyzed molecules (except ataciguat) will be removed from the central nervous system by the P-glycoprotein.

## 4. Riociguat in Pulmonary Hypertension

Pulmonary hypertension (PH) is a severe, progressive, and life-threatening disease in which the mean pulmonary artery pressure is equal to or exceeds 25 mmHg at rest [1,2]. PH can be a complication of common heart conditions, chronic lung disease, pulmonary embolism, or the result of an untreated congenital heart defect. Pulmonary arterial hypertension (PAH) is a rare form of pulmonary hypertension of unknown etiology. The disease is found in an average of 12–50 people/million. A larger group of patients with PH are those with chronic thromboembolic pulmonary hypertension (CTEPH). CTEPH is usually a complication of acute pulmonary embolism, which is about 20% of cases, that evolves into a chronic form characterized by vasoconstriction, vascular remodeling, the proliferation of smooth muscle and endothelial cells, and in situ thrombosis, ultimately causing an increase in pressure in the pulmonary artery and right heart chambers, with subsequent right heart failure [1,23,24]. 

The balance between vasodilatation and vasoconstriction plays a vital role in the proper functioning of the pulmonary circulation and, consequently, in optimal gas exchange and maintenance of the ventilation/perfusion (V′/O′) ratio balance. Vasodilatation is regulated by metabolic pathways dependent on nitric oxide and prostacyclin, while thromboxane A2 and endothelin, among others, are responsible for vasoconstriction. Under physiological conditions, these proteins are necessary to maintain peripheral vascular resistance and thus adequate blood pressure. In pulmonary hypertension, this balance is disturbed. In poorly ventilated areas, there are low concentrations of NO, resulting in vasoconstriction and preferentially ‘diverting’ blood to highly ventilated lung areas, where blood oxygenation is more efficient due to vasodilation. In more minor well-ventilated areas, in response to reduced bioavailability of NO and prostacyclin, endothelin production increases, causing chronic vasoconstriction and excessive proliferation of vascular endothelial cells (mitogenic effect), resulting in the remodeling within them [25].

The treatment of pulmonary hypertension is based on the action of three pathophysiological pathways that reduce pulmonary bed pressure: the prostacyclin pathway (prostaglandin I2 (PGI2)), the endothelin pathway, and the nitric oxide pathway. The action of endothelin antagonists, such as bosentan on the endothelin receptor on the vascular endothelial surface in both the pulmonary and peripheral circulation results in vasorelaxation. Prostacyclin produced by endothelial cells exhibit vasodilatory, antiproliferative, and antiplatelet properties. In PAH, prostacyclin production in the pulmonary arteries is impaired. Currently, prostacyclin analogs and synthetic prostacyclin have been used to treat PAH. These drugs, however, are also characterized by effects on the peripheral vasculature, often resulting in the discontinuation of therapy associated with poor drug tolerance. By preventing cGMP degradation through inhibiting phosphodiesterase 5 (PDE5), drugs, such as sildenafil, also cause vasodilatation, but their efficacy depends on the baseline NO concentration, meaning that in areas of the lung with low NO concentrations, their effect is limited [23,26,27,28,29]. 

So far, the drug mentioned above has not been used in thromboembolic pulmonary hypertension and pulmonary hypertension secondary to pulmonary disease. Due to the unsatisfactory treatment results with the available preparations, research on other drugs, including those using the NO-sGC-cGMP pathway, is being conducted all the time. [23] 

Riociguat, being a nitric-oxide-independent direct guanylyl cyclase stimulator, increases cGMP levels by the direct stimulation of sGC. The drug also increases sGC activity in the presence of low NO concentrations and increases the sensitivity of vascular wall cells to endogenous nitric oxide. It also provides vasodilation in lung areas with low ventilation, where previously used drugs had a poor effect. This phenomenon has been exploited in treating chronic thromboembolic pulmonary hypertension [30].

The first reports of the efficacy of riociguat in PH date back to 2010. At that time, Ghofrani et al. announced the results of a 12-week follow-up in patients treated with riociguat (42 patients with CTEPH and 33 with PAH) in which the drug improved exercise capacity and reduced symptoms of the disease while being well-tolerated [31]. In 2012, PATENT-1 (Pulmonary Arterial Hypertension Soluble Guanylate Cyclase-Stimulator Trial 1) results were announced. In this randomized, multicenter, double-blind study involving 443 adult patients with PAH, riociguat in doses of 0.5–2.5 mg was administered three times a day for 12 weeks. Compared to the placebo, improvements in performance parameters, i.e., increased distance in the 6 min walk test (6MWT), improved WHO functional class, decreased pulmonary vascular resistance (PVR), and decreased N-terminal prohormone brain natriuretic peptide (NT-proBNP) were observed after treatment with the drug [32].

The safety and efficacy of the prolonged use of riociguat were assessed in a group of 396 patients selected from the PATENT-1 study in an open-label trial. In this trial, conducted under the acronym PATENT-2, improvements in functional class and prolongation of 6MWT distance were still observed after one year of drug use, and good tolerability of riociguat treatment by the majority of patients was confirmed [33].

Due to the suspected additive effect of riociguat and phosphodiesterase five inhibitors, the PATENT PLUS study was performed. Patients treated with sildenafil for PAH were given riociguat for 12 weeks. There was no significant difference in hemodynamic parameters and exercise capacity improvements compared to the group treated with sildenafil alone [34].

In the CHEST-1 multicenter study among 261 patients with inoperable CTEPH, riociguat also showed improvements in 6MWT distance, WHO functional class, decreased pulmonary vascular resistance, and decreased NT-proBNP [35]. The long-term efficacy of riociguat was confirmed in the CHEST-2 trial, which included patients from the CHEST-1 trial, among others, and the improvements in 6MWT and the WHO functional class were maintained after one year of treatment [36].

Although the aforementioned studies achieved significant clinical improvement in riociguat-treated patients and decreased sPAP, mPAP, and PVR, they did not assess proper heart dysfunction parameters, which are vital prognostic factors in predicting survival in patients with PAH and CTEPH. Such an analysis was performed by Marra A. et al. among patients participating in the PATENT-1, PATENTplus, EAS, and CHEST trials, among 39 patients treated with riociguat (21 with PAH, 18 with CTEPH). The researchers continued the therapy for 12 months. They confirmed a significant decrease in correct ventricular dimensions at 3, 6, and 12 months of treatment, a decrease in correct atrial dimensions after 12 months, and an increase in TAPSE after 6 and 12 months of treatment in both PAH and CTEPH patients [37].

Furthermore, a study was conducted on patients with PH and left ventricular heart failure with LVEF <40% who received riociguat for 16 weeks (LEPHT study). Compared to the group not receiving the drug, there was a statistically significant improvement in hemodynamic parameters, such as systemic and pulmonary resistance and cardiac index; there was no significant changes in mPAP and PCWP; and trans-pulmonary gradient was found. The drug did not improve the NYHA performance class or slow cardiac function [38]. In contrast, the DILATE-1 study assessed the utility of the drug in diastolic heart failure. As in the LEPHT study, despite improvements in left ventricular parameters, mPAP and resting heart rate did not decrease [39].

An attempt to use riociguat in pulmonary hypertension secondary to chronic obstructive pulmonary disease was studied by Ghofrani et al. in 2015 in 22 patients with COPD and mPAP > 23 mmHg. After a single drug dose, they found a reduction in mean pulmonary artery pressure and a decrease in pulmonary vascular resistance (PVR) but no improvement in gas exchange parameters [40]. Two other studies in which riociguat was administered in patients with PH and primary lung disease failed due to poor tolerance of the drug and suspected increased mortality in this group of patients [41,42].

## 5. Vericiguat in Heart Failure

Among sGC activators, vericiguat has been used in clinical practice. In the SOCRATES-REDUCED trial, vericiguat was administered to patients with stable HFrEF. The drug did not worsen hemodynamic parameters and was well tolerated; however, after 12 weeks of treatment, there was no significant reduction in NT-proBNP or impact on the composite endpoint of HF hospitalization and cardiovascular mortality [43]. 

In another randomized trial (VICTORIA), in a cohort of 5050 heart failure patients with reduced LVEF <45% in NYHA class II-IV with elevated NT-proBNP values, 2524 patients received vericiguat given together with standard heart failure therapy for more than ten months. Inclusion criteria were either hospitalization for worsening heart failure in the last 3–6 months or the need for out-of-hospital use of intravenous diuretics for the same reason in the last three months. Patients taking intravenous positive inotropic drugs; treated with long-acting nitrates, other guanylyl cyclase stimulants (e.g., riociguat), or phosphodiesterase inhibitors; or using a left ventricular assist device were not included in the study. An 18-month follow-up was planned, but the median actual follow-up time was 10.8 months. After just three months of therapy, the authors noted its beneficial effects, which persisted until the end of the study. The results of the analyses showed that there was a significant (*p* = 0.019) 10 percent reduction in the risk of the composite endpoint of cardiovascular death or hospitalization for heart failure decompensation in the vericiguat-treated group compared with the placebo-treated group—mainly due to a decrease in the rate of the second component. The rate of adverse events (hypotonia and fainting) was similar in the vericiguat group and the placebo group (9.1 vs. 7.9 and 4.0 vs. 3.5, respectively) [44]. Subsequent post hoc analyses showed a lack of efficacy of the drug in the group of patients with the highest baseline NT-pro-BNP values > 8000 ng/L compared to patients with lower values. According to the investigators, despite optimal treatment, patients with a recent history of heart failure decompensation benefit most from vericiguat [45,46].

The feasibility of vericiguat in heart failure with preserved LV ejection fraction (HFpEF) was investigated in the randomized, double-blind SOCRATES-PRESERVED trial. Patients with EF >45% 4 weeks after hospitalization for heart failure or completion of treatment with intravenous diuretics were given the drug for 12 weeks. At the end of the trial, there were no significant changes in NT-proBNP levels, a reduction in the left atrial dimension compared with placebo. The lack of the expected effect may have been due to low doses of vericiguat [47]. Another randomized trial, VITALITY-HFpEF, followed 789 patients after an episode of decompensated HFpEF and randomized them into two groups—those receiving placebo and those receiving vericiguat at 10 and 15 mg/d. After 24 weeks of treatment, again in this study, there were no significant differences in performance improvement between the drug-treated and placebo groups [48]. 

Based on the aforementioned studies, the drug has found its way into clinical practice. Within the European Union, the drug is available from 27 July 2021. The indication for use in chronic heart failure with reduced left ventricular ejection fraction in patients who have recently developed a need for intravenous medication due to disease exacerbation and progression of symptoms. This indication can be found in the guidelines 2021 of the European Society of Cardiology for the treatment of heart failure (class of recommendations IIb) [49].

A similar application to vericiguat is likely to be cinaciguat, which can activate oxidized, dysfunctional, heme-group-free soluble guanylate cyclase, as well as in the case of reduced NO bioavailability and under the conditions of increased oxidative stress. Its use in clinical practice is still under investigation [3,5,7].

## 6. Pediatric Population

Treating heart failure in children is a huge challenge for the doctor. Management is difficult because a relatively small number of drugs are clinically tested in children, and children sometimes experience adverse effects related to growth. In the case of vericiguat, there is no evidence from clinical trials that would confirm the efficacy and safety of such treatment. The situation is complicated by the fact that in experimental studies on animals, vericiguat has been shown to affect the formation of properly functioning osteoclasts. Vericiguat modulated the differentiation of osteoclasts and showed dual effects on osteoclasts fusion and bone resorption in a dose-dependent manner. The results of molecular assays suggested that the dual regulatory effects of vericiguat on osteoclasts were mediated by the bidirectional activation of the IκB-α/NF-κB signaling pathway, thus influencing the formation of functional osteoclasts. These effects were not seen in adult animals. In these circumstances, the continuation of experimental studies is necessary to solve this problem before starting clinical trials in children [50].

## 7. Other Medicines

Praliciguat is another sGC activator undergoing clinical trials. The randomized CAPACITY HFpEF trial was conducted in a group of 196 patients with HFpEF for 12 weeks. There was no effect of the substance on improving ergospirometric parameters, 6MWT prolongation, or other biochemical and electrocardiographic parameters studied [51]. There are also ongoing studies on the use of cinaciguat in acute decompensated heart failure and oliciguat in the treatment of achalasia and sickle cell disease. The research on ataciguat in patients with aortic valve calcifications promises to be interesting. No papers summarizing these lines of research have been published.

## 8. Limitations

Activators and stimulators of guanylate cyclase, commonly called the guates family, were investigated in many different clinical trials. The differences in clinical trials make huge difficulties in clinical interpretation. Single results are clear, but currently the patient population itself is heterogeneous and the therapeutic procedures are different. Clinical trials with guates were conducted in groups of patients with heart failure with reduced systolic function, mildly reduced (previously mid-range systolic function, and preserved systolic function. Populations are different and, above all, concomitant pharmacotherapy is different and adjusted to clinical setting. Patients with heart failure with reduced ejection fraction are treated with a beta blocker, a converting enzyme inhibitor or a neprilysin inhibitor, a mineralocorticoid receptor antagonist, and sodium–glucose cotransporter-2 inhibitors. In case of fluid retention, a loop or thiazide diuretic is added. In the case of heart failure with preserved ejection fraction, only sodium–glucose cotransporter-2 inhibitors and diuretics have been shown to affect survival in clinical trials. Patients treated according to various recommendations took part in studies evaluating guates. In particular, a relatively small percentage of patients receiving sodium–glucose cotransporter-2 inhibitors and neprilysin inhibitors paradoxically made the effect of guates less pronounced in the absence of a synergistic effect with these groups of drugs. A similar situation occurred in the case of pulmonary arterial hypertension, but here the disease is more heterogeneous, and more studies have been published in similar populations. Currently, because of the same reason, it is impossible to compare such different populations in a meta-analysis, especially the in heart failure patient population. In this situation, it is particularly important to clearly define the role of guanylate cyclase activators and inhibitors. More clinical trials are needed in current and future clinical indications.

## 9. Perspective

Guanylate cyclases is widely distributed in almost all tissues. The guanylate cyclase pathway is one of the most common signaling pathways in the body. At the same time, differences in distribution do not ensure tissue or organ specificity for drugs acting through GCs. Reversing to normal or elevated activity has become an interesting therapeutic option.

Due to the way they act on the guates enzyme, the family can be divided into two groups of drugs—with a stimulating effect and with activating one. The first group is used in diseases, where it is only necessary to increase activity in the presence of gaseous mediators, such as NO or CO. Therefore, the clinical indication is pulmonary hypertension, erectile disfunction, etc. The treatment of heart failure, as well as sickle cell anemia or neuropathy, requires a significant increase in enzyme activity, usually without endogenous stimulation, hence the need to use activators. In this situation, only the reduction in the frequency of hospitalizations during treatment with vericiguat should be surprising. We believe that the explanation lies in the therapy used in these patients; more specifically the small proportion of patients treated with empagliflozin or dapagliflozin. The lack of a synergistic effect with other drugs affecting the activation of GCs thus limits the clinical effectiveness. Therefore, it can be expected that the continuation of clinical trials conducted among patients treated in accordance with current recommendations may completely change the role of guates in therapy. We can also expect an extension of the list of diseases in which the effect of exposure to GC will be a therapeutic effect.

Guates have a chance to become one of the leading groups of drugs, especially in the treatment of atherosclerotic diseases but also in neurology, hematology immunology, and many others.

## 10. Conclusions

The use of modulators of the GC function is currently the standard therapeutic procedure in PAH. In the case of GC activators, such as vericiguat, their importance in the treatment of heart failure can be expected to increase with further clinical trials. Due to the modulation of a widely spread link between guanylate cyclases, numerous new therapeutic indications for this family of drugs can be expected.

## Figures and Tables

**Figure 1 molecules-28-00861-f001:**
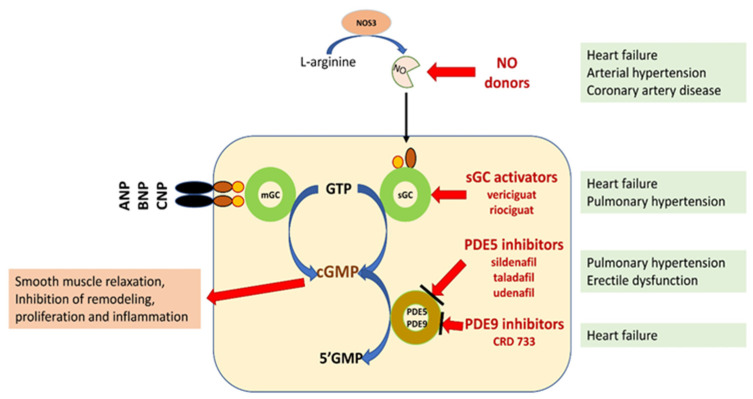
Schematic representation of the potential for pharmacological interference with the NO/sGC/cGMP pathway [3,5,6,7]. Explanation of abbreviations: sGC—soluble form of guanylyl cyclase; mGC—membrane form of guanylyl cyclase; NO—nitric oxide; eNOS—nitric oxide synthase; ANP, BNP, CMP—natriuretic peptide A, B or C; GMP 3′-5′—guanosine monophosphate; GTP 3′-5′—guanosine triphosphate; cGMP cyclic 3′-5′—guanosine monophosphate; CNGCs—cyclic nucleotide gated channels; PKG—protein kinase G; PDE phosphodiesterase.

**Figure 2 molecules-28-00861-f002:**
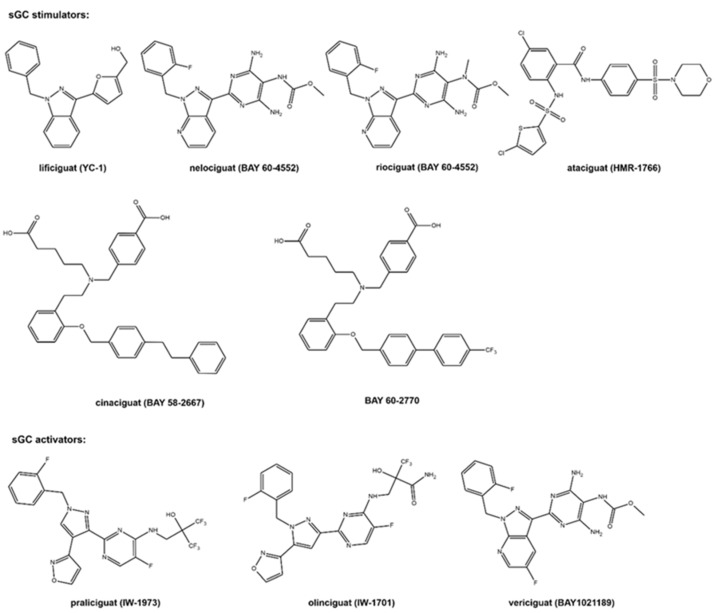
The molecular structure of the studied sCG compounds.

**Figure 3 molecules-28-00861-f003:**
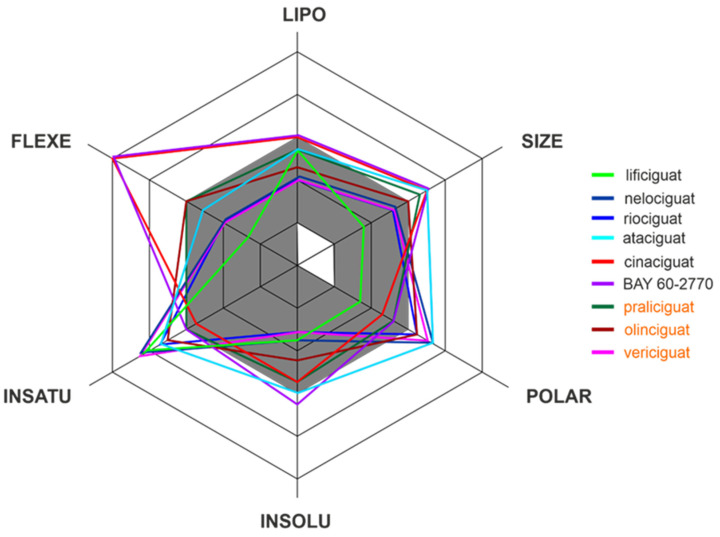
The bioavailability radar of the sCG activators/stimulators. The gray area represents the optimal range for each of the properties (lipophilicity: XLOGP3 between −0.7 and +5.0, size: MW between 150 and 500 g/mol, polarity: TPSA between 20 and 130 Å2, solubility: log S no higher than 6, saturation: fraction of carbons in the sp3 hybridization no less than 0.25, and flexibility: no more than 9 rotatable bonds). The compounds names are indicated in the different font colors, i.e., black corresponds to sGC stimulators and orange corresponds to sGC activators.

**Figure 4 molecules-28-00861-f004:**
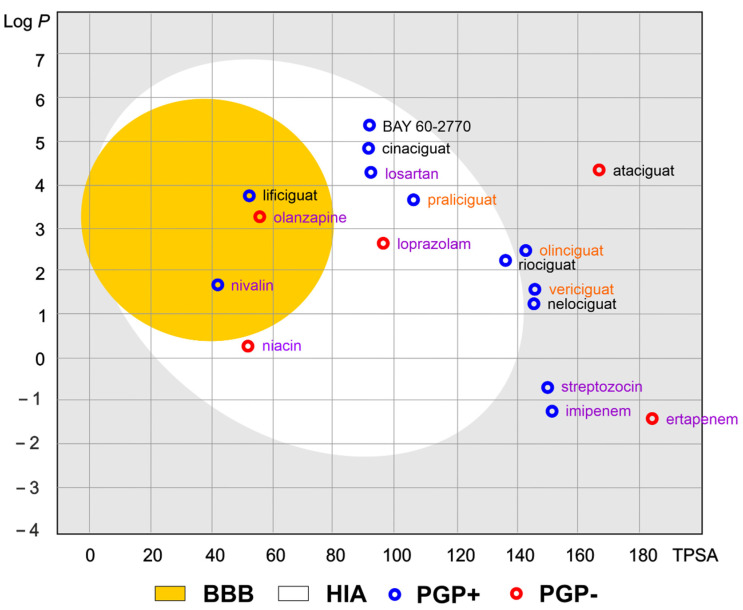
The Brain Or IntestinaL EstimatED (BOILED-Egg) graph compared sGC and selected commonly use drugs. The colored areas represent the optimal prediction range (above 93%) of penetration for brain (BBB, yellow) and gastrointestinal tract (HIA, white). The gray color indicates the area where penetration into the brain and gastrointestinal tract may occur but its probability is below the optimal value. The molecule predicted to be evaluated from the central nervous system by the P-glycoprotein (blue dots) or not (red dots). The compounds names are indicated in the different font colors, i.e., black corresponds to sGC stimulators, orange corresponds to sGC activators, and purple corresponds to selected drugs.

**Table 1 molecules-28-00861-t001:** Research directions using sCG activators in medicine [10,11,12].

Compound	Clinical Setting	Clinical Trial
riociguat	CTEPH	CHEST
riociguat	PAH, CTEPH, childhood PAH, erectile dysfunction	PATENT 2, CHEST 2, PATENT CHILD
vericiguat	HFrEF	SOCRATES-REDUCED
vericiguat	HF	SOCRATES -PRESERVED VICTORIA HFrEF, VITALITY HFpEF
praliciguat	HFpEF	CAPACITY -HFpEF
praliciguat	Diabetic neuropathy, sickle cell anemia, HF	STRONG SCD
oliciguat	Achalasia, sickle cell anemia	
cinaciguat	HF, acute decompensated HF	
Other compounds—potential clinical indications	Prostate hypertrophy, Reynaud’s syndrome in systemic sclerosis, cystic fibrosis, Duchenne muscular dystrophy, congenital bone fragility, non-alcoholic fatty liver disease, dementia, neuropathic pain, peripheral artery disease	

## Data Availability

Not applicable.

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
