# Peer review of "Soluble Guanylyl Cyclase Activators—Promising Therapeutic Option in the Pharmacotherapy of Heart Failure and Pulmonary Hypertension"

_molecules, 2023, doi:10.3390/molecules28020861_

Round 1

Reviewer 1 Report

The authors report an interesting review on effects of activators of soluble guanylyl cyclase in heart failure and pulmonary hypertension. Some reviews have been already published.

The major concern is that the authors report general findings. The methodology used do not allow to go more deeply on trials findings. Why meta analysis of the published studies was not chosen by the authors?

The authors should emphasize about new knowledge that the review can bring.

The chapter about children should be a separate chapter. 

Author Response

Response to Reviewer 1 Comments

Point 1 The authors report an interesting review on effects of activators of soluble guanylyl cyclase in heart failure and pulmonary hypertension. Some reviews have been already published.

Response 1 To date, many papers have been published on the assessment of guates. There are also reviews. In our work, we set the goal of presenting the relationship between chemical, pharmacological and clinical aspects. We also included the pediatric population, completely omitted in previous reviews.

The major concern is that the authors report general findings. The methodology used do not allow to go more deeply on trials findings. Why meta analysis of the published studies was not chosen by the authors?

Response 2. The decision to prepare a review and not a meta-analysis results from the clinical trials conducted so far. Currently, both the patient population itself is heterogeneous and the therapeutic procedure is different. Studies using guates were conducted in groups of patients with heart failure with reduced systolic function, mildly reduced (previously mid-range) and preserved systolic function. Populations are different and, above all, pharmacotherapy is different. Patients with HFrEF are treated with a beta-blocker, a converting enzyme inhibitor or a neprilysin inhibitor and a mineralocorticoid receptor antagonist and flozin. In case of overhydration, a loop or thiazide diuretic is added. In the case of HFpEF, only flozins and diuretics have been shown to affect survival in clinical trials. Patients treated according to various recommendations took part in studies evaluating guates (2016, 2018, 2021). In particular, a relatively small percentage of patients receiving flozins as neprilysin inhibitors paradoxically made the effect of guates less pronounced in the absence of a synergistic effect with these groups of drugs. a similar situation occurred in the case of pulmonary arterial hypertension, but here the disease is more heterogeneous and more studies have been published in similar populations. This factor also makes it impossible to compare such different populations in a meta-analysis. In the near future, studies on the use of guates in similar populations and indications will be published, and then we will certainly undertake the preparation of meta-analyses. Similar situation

The authors should emphasize about new knowledge that the review can bring.

Response 3. In reference to the answers to points 1 and 2 also, we have added a "Perspective" section. Thank you very much for this comment. Based on the facts presented in the previous chapters, we summarized the most important aspects, as well as the prospects for the development of this group of drugs that we expect.

  1. Perspective

Guanylate cyclases is widely distributed in almost all tissues. The guanylate cyclase pathway is one of the most common signaling pathways in the body. At the same time, differences in distribution do not ensure tissue or organ specificity for drugs acting through GCs. Reversing to normal or elevated activity has become an interesting therapeutic option.

Due to the way they act on the guates enzyme, the family can be divided into two groups of drugs - with a stimulating effect and with activating one. The first group is used in diseases where it is only necessary to increase activity in the presence of gaseous mediators such as NO or CO. Therefore, the clinical indication is pulmonary hypertension, erectile disfunction etc. Treatment of heart failure as well as sickle cell anemia or neuropathy requires a significant increase in enzyme activity, usually without endogenous stimulation, hence the need to use activators. In this situation, only the reduction in the frequency of hospitalizations during treatment with vericiguate should be surprising. We believe that the explanation lies in the therapy used in these patients, more specifically the small proportion of patients treated with empagliflozin or dapagliflozin. The lack of a synergistic effect with other drugs affecting the activation of GCs thus limits the clinical effectiveness. Therefore, it can be expected that the continuation of clinical trials conducted among patients treated in accordance with current recommendations may completely change the role of guates in therapy. We can also expect an extension of the list of diseases in which the effect of exposure to GC will be a therapeutic effect.

Guates have a chance to become one of the leading groups of drugs, especially in the treatment of atherosclerotic diseases but also in neurology, hematology immunology and many others.

The chapter about children should be a separate chapter. 

Response 4 “Pediatric” chapter is now a separate one. Thank You for suggestion.

Reviewer 2 Report

overall an excellent manuscript with good flow of thoughts and language. I would suggest adding a table with major learning points of this review and possible future research in this field.

Author Response

Response to Reviewer 2 Comments

Point 1 overall an excellent manuscript with good flow of thoughts and language. I would suggest adding a table with major learning points of this review and possible future research in this field.

Response 1. Thank you very much for your suggestion. We've added a "perspective" section. In reference to the answers to points 1 and 2 also, we have added a "Perspective" section. In this section, we have included comments on anticipated and current uses in relation to ongoing clinical trials. We also tried to list the most important aspects of guates (a kind of "take home message")

  1. Perspective

Guanylate cyclases is widely distributed in almost all tissues. The guanylate cyclase pathway is one of the most common signaling pathways in the body. At the same time, differences in distribution do not ensure tissue or organ specificity for drugs acting through GCs. Reversing to normal or elevated activity has become an interesting therapeutic option.

Due to the way they act on the guates enzyme, the family can be divided into two groups of drugs - with a stimulating effect and with activating one. The first group is used in diseases where it is only necessary to increase activity in the presence of gaseous mediators such as NO or CO. Therefore, the clinical indication is pulmonary hypertension, erectile disfunction etc. Treatment of heart failure as well as sickle cell anemia or neuropathy requires a significant increase in enzyme activity, usually without endogenous stimulation, hence the need to use activators. In this situation, only the reduction in the frequency of hospitalizations during treatment with vericiguate should be surprising. We believe that the explanation lies in the therapy used in these patients, more specifically the small proportion of patients treated with empagliflozin or dapagliflozin. The lack of a synergistic effect with other drugs affecting the activation of GCs thus limits the clinical effectiveness. Therefore, it can be expected that the continuation of clinical trials conducted among patients treated in accordance with current recommendations may completely change the role of guates in therapy.

Round 2

Reviewer 1 Report

The authors took into account my concerns. It will be an added value if authors can add some limitations such difficulty to perform meta analysis and emphasize the importance of setting RCTs.

Authors define the word "gates" in the perspectives chapter.
